# Remote Arrhythmia Detection for Eldercare in Malaysia

**DOI:** 10.3390/s21248197

**Published:** 2021-12-08

**Authors:** Kevin Thomas Chew, Valliappan Raman, Patrick Hang Hui Then

**Affiliations:** 1Faculty of Engineering, Computing and Science, Swinburne University of Technology Sarawak Campus, Kuching 93350, Sarawak, Malaysia; ktchew@swinburne.edu.my; 2Department of AI and DS, Coimbatore Institute of Technology, Coimbatore 641014, Tamil Nadu, India; valliappan@cit.edu.in

**Keywords:** arrhythmia, ECG classification, eldercare, electrocardiogram, RPM

## Abstract

Cardiovascular disease continues to be one of the most prevalent medical conditions in modern society, especially among elderly citizens. As the leading cause of deaths worldwide, further improvements to the early detection and prevention of these cardiovascular diseases is of the utmost importance for reducing the death toll. In particular, the remote and continuous monitoring of vital signs such as electrocardiograms are critical for improving the detection rates and speed of abnormalities while improving accessibility for elderly individuals. In this paper, we consider the design and deployment characteristics of a remote patient monitoring system for arrhythmia detection in elderly individuals. Thus, we developed a scalable system architecture to support remote streaming of ECG signals at near real-time. Additionally, a two-phase classification scheme is proposed to improve the performance of existing ECG classification algorithms. A prototype of the system was deployed at the Sarawak General Hospital, remotely collecting data from 27 unique patients. Evaluations indicate that the two-phase classification scheme improves algorithm performance when applied to the MIT-BIH Arrhythmia Database and the remotely collected single-lead ECG recordings.

## 1. Introduction

Despite rapid development of clinical practices and technologies, cardiovascular diseases continue to be a common concern in modern society. It remains the primary cause of death globally; representing 31% of all deaths that occurred worldwide in 2016 [1]. This trend is reflected in localized statistics collected by the Department of Statistics Malaysia. According to their 2019 press release, ischemic heart diseases are reported as the principal cause of Malaysian deaths in 2018 at 15.6%, rising from 13.9% recorded in 2017 [2]. The press release further indicates that ischemic heart diseases are the leading cause of death for individuals aged 41 and above.

In particular, cardiovascular diseases are a major concern for elderly citizens; a growing demographic that is more susceptible to developing these diseases and more likely to succumb to them [3,4]. In Malaysia, the proportion of citizens aged 60 and above has been steadily rising as a result of lower fertility rates, lower mortality rates, and increased life expectancy [5,6]. As the risk of disabilities and frailties increases with age [7,8], patient monitoring is crucial for the early detection of health deterioration and the onset of these diseases.

Through the use of electrocardiograms (ECG), medical staff are able to noninvasively monitor heart conditions based on electrical impulses generated by contractions of the heart muscles [9,10]. Interpretation of these signals have proven to be an effective method for detecting abnormal heart behavior [11,12], leading to widespread adoption in hospital settings. For other settings, advancements in technology have enabled the development of portable, affordable, and commercially available devices that are capable of capturing accurate ECG signals [13,14]. These developments open up new avenues through remote patient monitoring, allowing medical staff to monitor the conditions of individuals outside hospital grounds such as those in eldercare centers.

In this paper, we design and implement a medical platform for remote monitoring and automated arrhythmia detection in elderly individuals. To address the scalability and reliability requirements when managing a large number of remote data streams, a modular approach is proposed to allow for horizontal scaling of individual functionalities within the system. For automated heart arrhythmia detection, a two-phase classification scheme is introduced to improve the classification performance of an existing algorithm. Verification of the proposed classification scheme is performed on ECG recordings from the MIT-BIH Arrhythmia Database and single-lead ECG recordings collected by the prototype system.

The paper is organized as follows. Section 2 covers existing works in relation to remote patient monitoring and ECG classification. Section 3 outlines the system architecture and prototype setup. Next, Section 4 details the training and testing methodology for the ECG classification, as well as the two-phase classification scheme. Scalability testing and evaluation of the two-phase classification scheme are discussed in Section 5, followed by the conclusion in Section 6.

## 2. Existing Works

Existing research works primarily explore the concept of remote monitoring through the deployment of wireless devices and wireless sensor networks [15,16,17]. Khan et al. [15] designed a wireless body area network architecture around an IEEE802.15.4/Zigbee sensor network used for the transmission of sensor readings to a centralized hospital record computer via a service node. Similarly, González et al. [16] developed a remote monitoring platform for tracking the body temperature and heart rate of patients. The system consists of microcontrollers that process biometric signals from sensors attached to the patient, forwarding data to a nurse server via a local ad hoc wireless network. Another example is the implementation of a hand-held device by Pollolini et al. [17] for remotely monitoring heart failure patients. The device incorporates a two-lead electrocardiogram, a photoplethysmograph, and a bioimpedence meter, which transmit the data to a local internet gateway over low-power bluetooth connectivity.

A key trend among current developments is the primary focus on inpatient monitoring. As a result, these works do not address the scalability and reliability challenges of truly remote monitoring. Certain works such as the TELECARE system proposed by Szydło et al. [18] attempt to bridge the gap, supporting both inpatients and outpatients through the use of mobile devices. The TELECARE system presents a modular architecture encompassing data acquisition, data processing, and the web interface. However, the architecture is not designed around the scalability and reliability needs associated with continuous vital sign monitoring and data processing for anomaly detection.

In terms of automated arrhythmia detection, various approaches have been proposed for ECG classification using algorithms and machine learning techniques. Chazal et al. [19] proposed an automatic heartbeat classifier using ECG morphology and heartbeat interval features. The work utilizes classifier models based on linear discriminants for the final heartbeat classification. Meanwhile, Ye et al. [20] developed a different approach to arrhythmia classification that utilizes a combination of morphological and dynamic features. A support vector machine uses the extracted features to classify heartbeats into one of 15 heartbeat classes.

Recent works have seen increased focus on deep learning techniques and artificial intelligence such as the use of neural networks. Chauhan et al. [21] utilize a deep recurrent neural network with long short-term memory units in order to detect anomalous heartbeats. On the other hand, Kiranyaz et al. [22] propose a patient-specific heartbeat classifier based on a 1-D convolutional neural network architecture. Adaptive 1-D convolutional neural networks perform feature extraction and classification of the raw ECG signals for each patient. Complex deep learning architectures have also been explored, such as the model developed by Qiu et al. [23] containing a backbone network for generating multiscale semantic and morphological feature maps.

When reviewing the works covering ECG classification, no clear consensus is reached on the right methodology to employ during the data preparation phase. Many existing works do not adhere to the Association for the Advancement of Medical Instrumentation (AAMI) recommendations, resulting in performance results that are higher than expected. For example, the works proposed by Ye et al. and Song et al. have been shown to produce lower performance results when the data are prepared following AAMI recommendations [24]. Due to the lack of standardization, comparisons of performance results between these works may not be accurate or representative of the actual classification performance.

Additionally, some works adhere to the AAMI recommendations but do not implement the interpatient scheme during data partitioning for the training and test datasets. When following the interpatient scheme, data from each patient can only appear in either the training dataset or test dataset to avoid learning bias. While the works proposed by Kiranyaz et al. and Qiu et al. adhere to the AAMI recommendations for excluding paced heartbeats and classifying heartbeats into five recommended heartbeat types, data from each patient are present in both the training and test datasets which can result in inflated performance results.

## 3. System Architecture

A high-level overview of the system architecture and its primary components are demonstrated in Figure 1. The system components can be divided into two major sections: (1) end user and (2) backend. The end user section consists of the sensory devices such as patches, the web client, and the mobile client. These components act as the primary interface for interacting with the system and are geographically distributed across diverse locations. The backend section consists of the reverse proxy, web API, ECG classifier API, and the database. These software-based components are hosted using cloud services and provide the core functionality of the system.

Data within the system are primarily generated and consumed on the end user side. All requests to access or store data are routed through the reverse proxy. The reverse proxy acts as a gateway to the backend functionality, managing all incoming requests and performing load balancing when multiple instances of backend components are deployed. The web API is responsible for interpreting incoming requests and performing the appropriate actions in response. For example, the web API interacts with the database to perform data access and storage. Additionally, the web API interacts with the ECG classifier API in order to detect abnormal heart arrhythmias in collected ECG signals and generate alerts.

### 3.1. Module System

The web API is designed around a module system. Each module provides all necessary components such as API routes and database mappings for a specific functionality. This approach allows for multiple deployments with varying feature sets as well as future extension through the inclusion of new modules. For example, new monitoring devices and the necessary functionality for handling its data can be added to the existing system, rather than requiring a complete rework of the system. The system includes two modules: (1) temperature monitoring, and (2) ECG monitoring. The web and mobile clients can provide access to all available modules or a subset as shown in Figure 2.

The temperature and ECG monitoring modules provide functionality for remotely monitoring the body temperature, heart rate, respiratory rate, and electrocardiogram readings of a patient. The patient’s vital signs are measured using a VV200 temperature patch and VV330 ECG patch from VivaLNK. As demonstrated in Figure 3, the patches connect and transmit data to the mobile client using bluetooth low energy (BLE). The temperature patch transmits data every 15 s while the ECG patch transmits data once every second.

Due to the high frequency of data, the mobile client acts as a buffer for incoming data using a local SQLite database. The data are transmitted to the web API in batches every 15 min by default, with varying frequencies depending on the immediate situation as indicated in Table 1.

### 3.2. Alert Generation

When certain events such as abnormal heart rhythms occur, the system generates alerts in order to inform medical practitioners of a potential problem. The alert generation process uses a job queue to keep track of incoming data that has not been processed for abnormalities. New sections of data are marked automatically and added to the job queue. As shown in Figure 4, the marking process also works for data that are received out of order from collection. The marked sections are wider than the unprocessed areas to support alert generation rules that may require the additional data for context.

Each entry in the job queue and its associated marked section of data are loaded by the web API. For the detection of heart arrhythmias, the web API forwards the loaded ECG signals to the ECG classifier API for processing. After classification, the web API generates alerts for any abnormal heartbeats that are detected. These alerts are saved to the database for viewing by the web and mobile clients.

### 3.3. Scalability

In contrast to systems designed for on-site deployments, remote monitoring systems are not constrained by physical limitations and may support upwards of hundreds to thousands of monitoring devices that are continuously producing data. These devices may not be positioned in a fixed location and may transmit data at inconsistent timings, requiring the system to adapt to variable loads with random spikes of activity.

As indicated in Figure 5, deployments of remote monitoring systems have greater flexibility in terms of geographical location and quantity of monitoring devices. These systems can theoretically support an unlimited number of users from various locations and organizations. The primary limitation to the maximum number of users is the capabilities of the server software in processing the incoming requests.

In the proposed system, the three major areas that affect overall scalability are the maximum number of concurrent requests, processing speed of these requests, as well as data access and storage capabilities. The two components responsible for handling all incoming requests are the web API and ECG classifier API. On the other hand, the database is the sole component that affects the data access and storage capabilities.

For each of these components, a single instance of the software can only handle a limited amount of load. The maximum load that can be handled is dictated by the available system resources of the host server. In order to support a higher level of load with a single instance of the software, vertical scaling can be done by upgrading the server hardware. However, the single instance architecture requires expensive server upgrades and cannot scale indefinitely due to physical and hardware constraints.

In order to address the limitations of vertical scaling, the proposed system is designed to also support horizontal scaling. Each component in the system can have one or more software instances running concurrently to service incoming requests. Figure 6 demonstrates the dynamic component scaling that can be adopted depending on the expected server load. For small-scale use cases, a single instance of each component can be used. As the number of users increases, additional servers running software instances can be spun up to handle the increased load. The proxy acts as a load balancer, distributing incoming requests evenly among all running instances.

While horizontal scaling provides greater redundancy and flexibility, the scale of operations ultimately determines the overall cost and performance efficiency. At smaller scales, horizontal scaling is inefficient due to the increased overhead and complexity of managing multiple software instances. Instead, the system can initially be scaled vertically on a single server to avoid the time and monetary costs of managing multiple servers. Once the number of users exceeds the processing capabilities of the single server instance, horizontal scaling of individual components can be performed based on the specific usage patterns and bottlenecks encountered.

### 3.4. Prototype Setup

A prototype setup was developed and deployed for use by the Sarawak General Hospital in Malaysia to evaluate the remote monitoring capabilities of the proposed system. The system was tested on a total of 27 unique patients, eight of which are stroke patients. The VV330 ECG patches from VivaLNK were used to monitor the heart rate, respiratory rate, and ECG signals of the patients for a period between 24 and 72 h. With the exception of the eight stroke patients, all other users were remotely monitored outside the hospital premises.

Android phones were used as the mobile client due to their lower cost and wide availability. A mobile client application was developed using the Kotlin programming language, with native Android features such as SQLite being used to implement data caching. For communication with the ECG patches, a software development kit (SDK) created by VivaLNK was used to receive incoming data from the ECG patches over a Bluetooth connection. In order to facilitate the remote monitoring process, a web client was also developed using standard web tools such as the hypertext markup language (HTML), cascading style sheets (CSS), and the Javascript programming language (JS).

For the backend, a single instance of each component is used as it was sufficient for the number of patients being monitored. The Nginx web server is used as a reverse proxy for the system. The web API and all associated modules are implemented in the Javascript programming language using the Node.js platform. For the ECG classifier API, the Python programming language is used as the selected ECG classification algorithm was originally developed in the same language. Finally, all data access and storage in the system is handled by a PostgreSQL database with a time-series extension, providing the structure of a relational database and the flexibility of a non-relational database.

## 4. Methodology

### 4.1. ECG Classification Algorithm

For the automatic classification of single-lead ECG recordings, the classification algorithm proposed by Mahajan et al. [25] was chosen. The proposed algorithm was selected as its implementation is open-sourced and the pretrained model weights used in the initial study are publicly available, allowing for direct comparisons between derivative works that employ similar methodologies.

The ECG classification was performed using a random forest classifier. The classifier consisted of an ensemble of 220 decision trees used to categorize four classes of ECG rhythms: normal, atrial fibrillation (AF), other AF, and noisy. Each decision split of the random forest classifier operates on a random subset of features, increasing the distinctiveness of outputs among the decision trees.

### 4.2. Dataset Preparation

#### 4.2.1. Data Selection

The MIT-BIH ECG Arrhythmia Database [26] was used as the primary data source during dataset preparation. Recommendations from the AAMI were used for the preparation of the training and test datasets. A total of 44 recordings were selected for use in both datasets. The four records that contain paced beats (102, 104, 107, 217) were excluded. As recommended by the AAMI, the MIT-BIH heartbeat types were combined into five class labels: N, S, V, F, and Q. The mapping of the original annotations to their final labels is shown in Table 2.

The training and test datasets were derived following the interpatient scheme to avoid bias in the classification performance. Under this scheme, the recordings were divided between the two datasets such that heartbeats from one patient were not used simultaneously for both the training and testing of the ECG classifier. Thus, the training dataset was composed of heartbeats from recordings 101, 106, 108, 109, 112, 114, 115, 116, 118, 119, 122, 124, 201, 203, 205, 207, 208, 209, 215, 220, 223, and 230. Meanwhile, the test dataset was composed of heartbeats from recordings 100, 103, 105, 111, 113, 117, 121, 123, 200, 202, 210, 212, 213, 214, 219, 221, 222, 228, 231, 232, 233, and 234.

The PhysioNet Computing in Cardiology Challenge 2017 [27] was used as a supplementary source of data for the training and test datasets. In particular, there were a very limited number of heartbeats under the unclassifiable beats category in the MIT-BIH dataset. Due to the exclusion of MIT-BIH records containing paced beats, the Q class only contained a total of 33 heartbeats after annotation mapping. Thus, noisy samples from the PhysioNet challenge data were used in order to reduce the imbalance between the classes.

As with the MIT-BIH records, the challenge data were divided such that each recording was only used once in either the training dataset or the test dataset. Starting from the first recording labelled A00022, each subsequent record was assigned to either the training or test dataset in an alternating fashion. Under this scheme, record A00022 was first assigned to the training dataset, followed by record A00034 being assigned to the test dataset, and so forth. Table 3 briefly demonstrates the concept used for dividing the records.

#### 4.2.2. Data Preparation

Initially, all ECG signals from the PhysioNet challenge data were upsampled from 300 Hz to 360 Hz using the Fourier resampling technique in order to match the ECG signals in the MIT-BIH database. Next, two median filters were applied to all ECG signals with a width of 200 ms and 600 ms in order to remove baseline wander and reduce signal spikes caused by noise. After filtering, amplitude normalization was performed on the resultant signals by rescaling to fit within a range between 0 and 1. The normalized signal eliminated potential biases from differences in the amplitude between different ECG signals.

After signal preprocessing, each successive PQRST wave present in the ECG signal was extracted as an individual heartbeat as demonstrated in Figure 7. Each heartbeat was extracted from the ECG signal using a window of 600 ms size, resulting in the extraction of 216 samples per heartbeat. Each window contained 108 samples from before the R-peak and 108 samples from after the R-peak. Each extracted heartbeat was assigned their original label based on the heartbeat annotations provided in the MIT-BIH database, while heartbeats extracted from the PhysioNet challenge data were assigned the Q class representing noisy readings.

### 4.3. Two-Phase Classification Scheme

#### 4.3.1. Motivation

Identifying abnormal heartbeats and their specific types from an ECG recording is a complex task. As demonstrated in existing studies, classification performance is inconsistent and varies based on the specific type of heartbeat that is being classified. In addition, certain applications such as anomaly detection and alert systems may only require a distinction between normal and abnormal heartbeats without requiring the specific abnormality type. In these scenarios, the additional computation performed when identifying the specific heartbeat abnormality type is unnecessary.

Therefore, the two-phase classification scheme was proposed with the objective of developing a technique for improving the performance of existing ECG classification algorithms. The classification scheme was designed with the intention of being algorithm-agnostic, allowing it to be replicated for other ECG classification algorithms besides the one discussed in this paper. Additionally, this approach allows for the combination of different classification algorithms that are specialized for the classification problem faced at each of the two stages.

Decomposing the classification process into two distinct stages provides several tangible benefits. The classification complexity at each of the two stages is reduced compared to the original approach of only classifying once. Each stage of the classification process places greater emphasis on addressing a specific problem. The first stage is focused on determining whether a given heartbeat is considered normal or abnormal, while the second stage focuses on identifying the specific abnormality type for heartbeats that are not categorized as normal.

#### 4.3.2. ECG Classification Process

The two-phase classification scheme was designed based on the classification approach commonly employed by cardiologists. Firstly, the baseline characteristics of a normal heartbeat is established for a given patient. Once the baseline is established, abnormal heartbeats that are present in the ECG signal are identified as heartbeats that deviate from the norm. Finally, the abnormal heartbeats are classified into specific categories based on their characteristics.

In order to perform the two-phase classification, a composite model consisting of two separately trained models is used. Figure 8 illustrates the flow of operations during the two-phase classification process. The first model classifies heartbeats into two classes, distinguishing between normal heartbeats and abnormal heartbeats. In this case, any unclassifiable signal or heartbeats that cannot be categorized as the N label are considered abnormal and classified as the A label. If a heartbeat is classified as abnormal by the first model, the second model is used to determine the type of abnormal heartbeat. The heartbeat is classified into one of the four abnormal heartbeat types as either S, V, F, or Q.

### 4.4. Training and Testing Procedure

In order to assess the classifier performance during training and testing, three metrics are employed: accuracy, sensitivity, and specificity. When evaluating the performance of the heartbeat classifier, the overall results for each of the outlined performance metrics is calculated. Additionally, each performance metric is also calculated on a per-class basis for each heartbeat classification label.

The terms TP, TN, FP, and FN denote true positive, true negative, false positive, and false negative respectively. True positive and true negative represent the number of samples that are accurately classified as belonging or not belonging to a given class. Conversely, false positive represents the number of samples that are misclassified as a given class that they do not belong to. Finally, false negative represents the number of samples that belong to a given class but are misclassified as a different class.

Equation (1) denotes the accuracy metric, measuring the total number of heartbeats that are correctly classified across all classes. Equation (2) denotes the sensitivity metric, measuring the proportion of heartbeats that are correctly identified as positive for a given class. Equation (3) denotes the specificity metric, measuring the proportion of heartbeats that are correctly identified as negative for a given class.
(1)$$\mathrm{Accuracy}=\frac{\mathrm{TP}+\mathrm{TN}}{\mathrm{TP}+\mathrm{TN}+\mathrm{FP}+\mathrm{FN}}$$
(2)$$\mathrm{Sensitivity}=\frac{\mathrm{TP}}{\mathrm{TP}+\mathrm{FN}}$$
(3)$$\mathrm{Specificity}=\frac{\mathrm{TN}}{\mathrm{TN}+\mathrm{FP}}$$

Initially, the random forest classification model based on the selected ECG algorithm is developed using the training dataset. An initial population of 10 individuals is randomly generated. Each individual in the population consists of an ensemble of bagged decision trees with an estimated 220 learners. All training of the initialized random forest model is performed exclusively on the training dataset to avoid introducing biases and discrepancies in the final classification performance. Once the model is fully trained, its classification performance against unseen data is evaluated using the test dataset. The accuracy, sensitivity, and specificity metrics are calculated in order to benchmark the final performance of the model.

After the baseline evaluation is performed, a two-phase classification scheme is applied to the same ECG classification algorithm. The proposed method is derived based on the approach typically employed by cardiologists when identifying abnormalities in ECG recordings. For the two-phase classification scheme, modified datasets are created for the training and evaluation of a composite model. Finally, the results of the baseline evaluation and the composite model are compared to determine whether there are measurable improvements in the evaluation metrics. Evaluation is also performed against a single-lead ECG dataset collected by the prototype system. The single-lead ECG signals were prepared following the outlined data preparation methodology, with the heartbeats being hand-labelled by a certified cardiologist from Sarawak General Hospital (SGH).

## 5. Evaluation & Results

### 5.1. Scalability Evaluation

In order to validate the benefits of horizontal scaling, test deployments were performed with a dynamic number of web API instances. The web API was chosen for evaluation as it performs the primary processing and data management tasks within the system. Additionally, the performance characteristics of the employed database software when sharding horizontally have been thoroughly tested and documented. Figure 9 illustrates the deployment architecture used for the evaluation.

A total of eight separate deployment configurations were used for testing. Each deployment had a fixed number of web API instances ranging between 1 to 8 instances. In order to simulate realistic cloud deployment scenarios, each web API instance was isolated using Linux containers allocated with 1 dedicated CPU and 512MB of system memory. The reverse proxy distributes incoming requests evenly between the running instances. A HTTP benchmarking tool was used to stress test the deployments and collect performance metrics.

The horizontal scaling was evaluated using three performance metrics: requests per second, latency, and errors. The requests per second metric refers to the number of HTTP requests that were successfully processed by the system. Latency was computed as the time from the sending of the first byte of the request to the time the complete response was received. Finally, the errors were computed as the number of requests that receive an erroneous response or requests that were not processed due to overloading of the system by the benchmarking tool.

Figure 10 demonstrates that the number of requests successfully processed per second increases with the number of deployed instances, plateauing between five and eight instances due to the single reverse proxy and database in the test environment. On the other hand, Figure 11 indicates that raising the number of deployed instances steadily decreases both the latency and number of erroneous requests. As each instance can only process a limited number of requests simultaneously, spreading the load across multiple instances provided greater parallel processing of incoming requests.

### 5.2. Prototype Deployment

Initial usability evaluation of the prototype system was performed at the Sarawak General Hospital in Malaysia. In total, the system was used on 27 unique patients to identify the effects of using a remote monitoring system. Each patient was outfitted with a VV330 ECG patch for continuous monitoring over a period between 24 to 72 h.

The application of wireless ECG patches enhanced patient compliance to prolonged cardiac monitoring in several ways. The continuous monitoring mechanism reduced the burdens typically associated with patient-initiated intermittent monitoring strategies using conventional ECG, Holter and novel mobile devices. Examples of these burdens include failure to adhere to ECG monitoring intervals and improper usage of the ECG monitoring devices.

Additionally, the wireless and waterproof features of the ECG patches resulted in tolerable comfort levels during the prolonged monitoring periods above 48 h. As the device is small and lightweight, patients were able to shower throughout the monitoring period. The absence of wired skin electrodes also reduced motion artefacts during ambulation and other daily activities.

### 5.3. Baseline Evaluation

For the baseline evaluation, the classification algorithm proposed by Mahajan et al. [25] was used as the reference specification for the implementation and training of the classification model. Figure 12 presents the confusion matrix after classification of the heartbeats in the test dataset. For each true label in the confusion matrix, the number of predicted heartbeats was normalized against the total number of heartbeats in the category and rounded to three digits.

From the confusion matrix, two extremes in terms of predictive performance are demonstrated. The N, V, and Q labels are shown to have a high percentage of their samples that are correctly classified by the model. The percentage of correctly classified samples for the N, V, and Q labels are 88.9%, 89%, and 97.7% respectively. On the other hand, the model is shown to perform poorly when classifying samples that belong to the S and F labels. For the S and F labels, the percentage of correctly classified samples are 0% and 3.6% respectively.

The model exhibited a specific characteristic when classifying samples that belong to the S and F labels. When examining the classification results for these labels, it is clearly shown that the majority of the samples belonging to these labels were misclassified as the N label. A total of 95.1% of the samples belonging to the S label were misclassified as the N label, while 77.8% of the samples belonging to the F label were misclassified as the N label. As for the remaining samples under the S and F labels, the majority of the misclassification occurs when the samples were mislabeled as the V label.

Generally, the performance metrics of the model paint a positive picture as the model achieved an accuracy score of 89.481% and a specificity score of 92.322%. However, the poor performance of the model when distinguishing heartbeats under the S and F labels resulted in a sensitivity score of 55.827%. The results from both Figure 12 and Table 4 demonstrate a clear trend in the classification performance of the model. The model performed well when classifying heartbeats that belong to the N, V, and Q labels. However, the model was unable to accurately distinguish heartbeats that belong to the S and F labels. The majority of these heartbeats were either misclassified under the N label or the V label.

### 5.4. Two-Phase Classification Scheme

#### 5.4.1. Models A and B

Both model A and model B are trained and tested using all available data within their respective training datasets. Model A was trained on a total of 17,190 heartbeats while model B was trained on a total of 8148 heartbeats. For the evaluation of the two models after training, a total of 17,033 heartbeats were used for testing model A while a total of 8268 heartbeats were used for testing model B.

Figure 13 presents the confusion matrix for model A after classification of the heartbeats in the test dataset. From the confusion matrix, the model is shown to perform better at identifying normal heartbeats as compared to abnormal heartbeats. The model correctly classified 88.9% of the normal heartbeats, while the remaining 11.1% were misclassified as abnormal heartbeats. The model demonstrated a lower performance when classifying abnormal heartbeats as it classified 76.2% of the abnormal heartbeats correctly, while the remaining 23.8% were misclassified as normal heartbeats. The performance metrics for model A were derived from the confusion matrix and presented in Table 5.

The classification results for model B on the test dataset are summarized in Figure 14. The confusion matrix indicates that the model performed well when classifying heartbeat samples under the S, V, and Q labels. The percentage of correctly classified heartbeats under the S, V, and Q labels are 87.5%, 92%, and 96.6% respectively. For the heartbeats under the F label, the model was only able to classify 59% of the samples correctly while the remaining samples were primarily misclassified under the S and V labels. The performance metrics for model B were derived from the confusion matrix and presented in Table 6.

#### 5.4.2. Composite Model

After both model A and model B were trained, the two model instances were combined in order to form the composite model. The performance of the composite model was evaluated using the same dataset as the baseline evaluation, containing a total of 12,648 heartbeats. The heartbeat classification results on the test dataset are presented in Figure 15.

Referencing the confusion matrix, the model is shown to have better predictive performance when classifying heartbeats that belong to specific labels. The model was able to classify a high percentage of the samples that belong to the N, V, and Q labels. Heartbeat samples under these labels were correctly classified at a rate of 89.8%, 82.5%, and 91.9% respectively.

In comparison, the model performed poorly when classifying samples that belong to the S and F label. For the S label, the model was only able to correctly classify 45.6% of the heartbeat samples. The majority of the remaining samples under the S label were misclassified as the N label. A similar trend is apparent when examining the F label, as the model was unable to correctly classify any of the samples correctly and the majority of the misclassified samples were predicted to be under the N label.

Table 7 contains the performance metrics of the composite model after collation of the heartbeat predictions. When considering the accuracy and specificity metrics, the composite model is shown to perform well in terms of predictive performance. Overall, the composite model achieved an accuracy of 91.385% and a specificity of 93.747%. However, the model achieved a lower overall result for the sensitivity metric at 61.959%. This can be attributed to the poorer classification performance when handling heartbeat samples that belong to the S and F labels.

### 5.5. Evaluation Results

During the baseline evaluation, a key trend can be identified from the classification results. When adapted to work on the MIT-BIH Arrhythmia Database, the selected ECG classification algorithm demonstrated polarizing performance depending on the type of heartbeat being classified. The baseline model performed well when classifying heartbeats that belong to the N, V, and Q labels. However, the model was shown to be unable to correctly identify heartbeats that belong to the S and F labels.

The same trend was present to a lesser degree when examining the classification results of the composite model. The composite model also performed well when classifying heartbeats under the N, V, and Q labels. Additionally, the composite model was also unable to identify heartbeats that belong to the F label. The key distinction was the marked improvement in classification performance for heartbeats under the S label, resulting in greater overall metrics as shown in Table 8.

The preparation process of the composite model provides greater insights into the predictive capabilities of the selected ECG algorithm. The algorithm is found to perform well when classifying abnormal heartbeat types, but faced difficulties in distinguishing between normal and abnormal heartbeats. In Table 6, it is shown that model B, which was trained specifically for identifying abnormal heartbeat types, was able to perform well for the S, V, and Q labels. Additionally, the model was able to correctly classify 59% of the heartbeats under the F label as shown previously in Figure 14.

In contrast, model A, which was trained specifically for distinguishing between normal and abnormal heartbeats, was only able to achieve lower classification performance results as shown in Table 5. While the overall results remained relatively high, there was a clear reduction in the predictive performance metrics when compared to the overall results for model B. This coincides with the results collected during the baseline evaluation, as the majority of the heartbeats misclassified by the baseline model were predicted to be normal heartbeats.

### 5.6. Evaluation on Single-Lead ECG Signals

In order to validate the classification performance against single-lead ECG signals, an additional dataset is prepared using the data collected by the prototype system at the Sarawak General Hospital in Malaysia. A total of eight stroke patients were monitored remotely using VV330 ECG patches from VivaLNK for a period between 24 and 72 h. From the collected ECG recordings, a subset containing 5 h of ECG strips was extracted and the heartbeats were hand-labelled by a certified cardiologist.

The extracted ECG signals were preprocessed following the previously established data preparation process. As the ECG patches record the ECG signals at 128 Hz, the signals were first upsampled from 128 Hz to 360 Hz using the Fourier resampling technique. Two median filters of 200 ms and 600 ms width were used to remove baseline wander from the upsampled signals, followed by amplitude normalization to a range between 0 and 1. Finally, the heartbeats were segmented using a window centered around the R-peaks with a size of 600 ms. Table 9 provides the total number of extracted heartbeats under each classification label.

The classification results for the baseline model and composite model are shown in Figure 16 and Figure 17. The baseline model performed poorly when classifying all heartbeats except those belonging to the V label, achieving a rate of 94.8%. From the confusion matrix in Figure 16, the baseline model is shown to classify the majority of the heartbeats under the V label, resulting in a performance imbalance for the other labels.

In comparison, the composite model demonstrates better performance when classifying heartbeats belonging to the N and V labels. Figure 17 demonstrates that the composite model correctly identified these heartbeats at a rate of 83.2% and 97.8% respectively. However, it was also unable to correctly identify heartbeats under the S and F labels. For the Q label, classification performance remains low but is improved from 35.1% to 43.5% when compared to the baseline model.

Table 10 and Table 11 provide the classification performance metrics of the baseline model and composite model on the single-lead ECG signals. The overall metrics for the two models are summarized in Table 12. From the results, both the baseline model and the composite model are shown to perform poorly when classifying heartbeats that belong to the S and F labels. Additionally, both models demonstrate a lower performance on the single-lead ECG signals when classifying heartbeats under the Q label.

When comparing the two models directly, the composite model generally demonstrates a higher performance across all metrics. Exceptional cases where the baseline model had higher performance results were the accuracy and specificity for the F label, as well as the specificity for the N label. When evaluating the overall metrics, the composite model is shown to be the better classifier as it outperformed the baseline model across all metrics.

## 6. Conclusions

To summarize, this research has successfully proposed and evaluated a scalable remote patient monitoring platform with automatic heart arrhythmia detection. A modular approach was employed for both the high-level architecture and the software design, allowing for greater horizontal scaling as the number of data streams increases. A prototype was developed and deployed for use by the Sarawak General Hospital in Malaysia, with a total of 27 unique patients being remotely monitored between 24 and 72 h.

For automated heart arrhythmia detection, a two-phase classification scheme was proposed in order to improve the classification performance of existing algorithms. An existing random forest classifier algorithm was retrained and reevaluated based on the proposed methodology. Evaluation of the experimental results indicate that the composite model from the two-phase classification scheme had a higher overall classification performance when compared to the original algorithm. The same results were reflected when classifying single-lead ECG recordings collected by the prototype system.

## Figures and Tables

**Figure 1 sensors-21-08197-f001:**
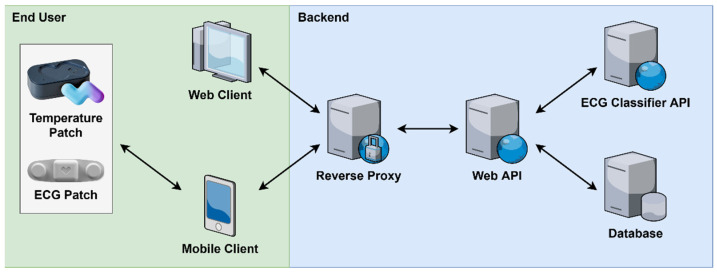
Overview of system architecture.

**Figure 2 sensors-21-08197-f002:**
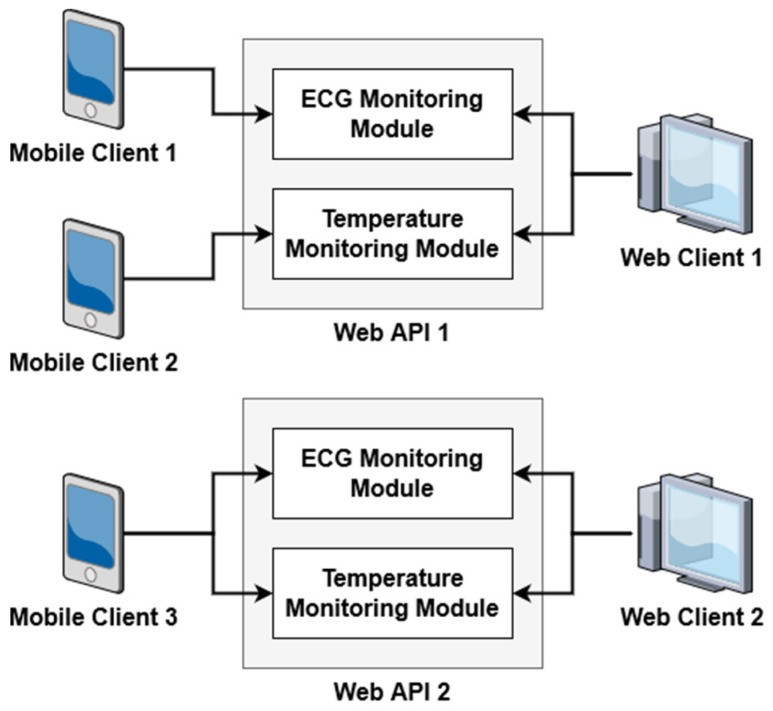
Overview of module system.

**Figure 3 sensors-21-08197-f003:**
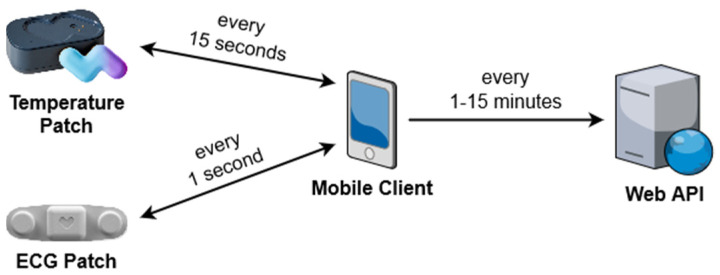
Communication between patches, mobile device, and server.

**Figure 4 sensors-21-08197-f004:**
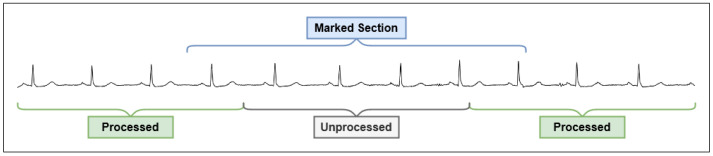
Marking of data sections that have not been processed.

**Figure 5 sensors-21-08197-f005:**
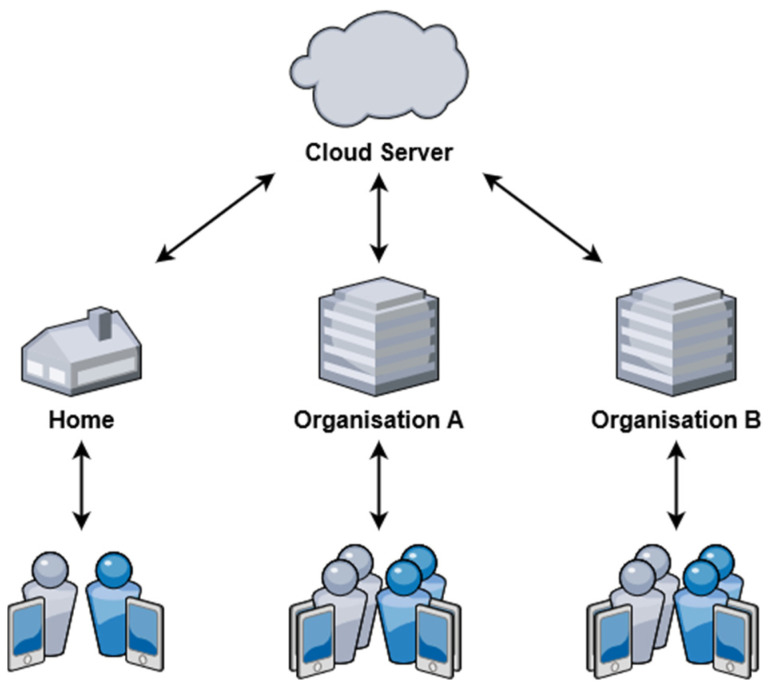
Various end users of a remote monitoring system.

**Figure 6 sensors-21-08197-f006:**
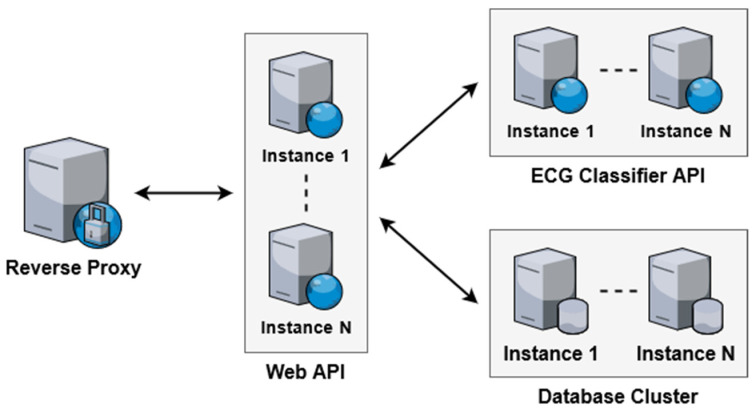
Dynamic scaling using multiple component instances.

**Figure 7 sensors-21-08197-f007:**
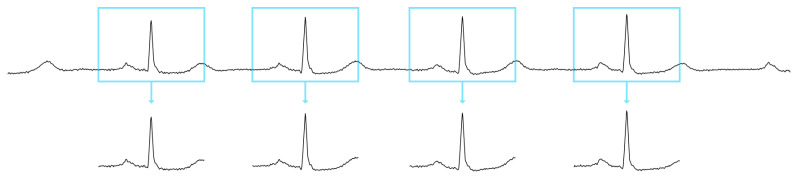
ECG signal snippet and windows containing extracted heartbeats.

**Figure 8 sensors-21-08197-f008:**
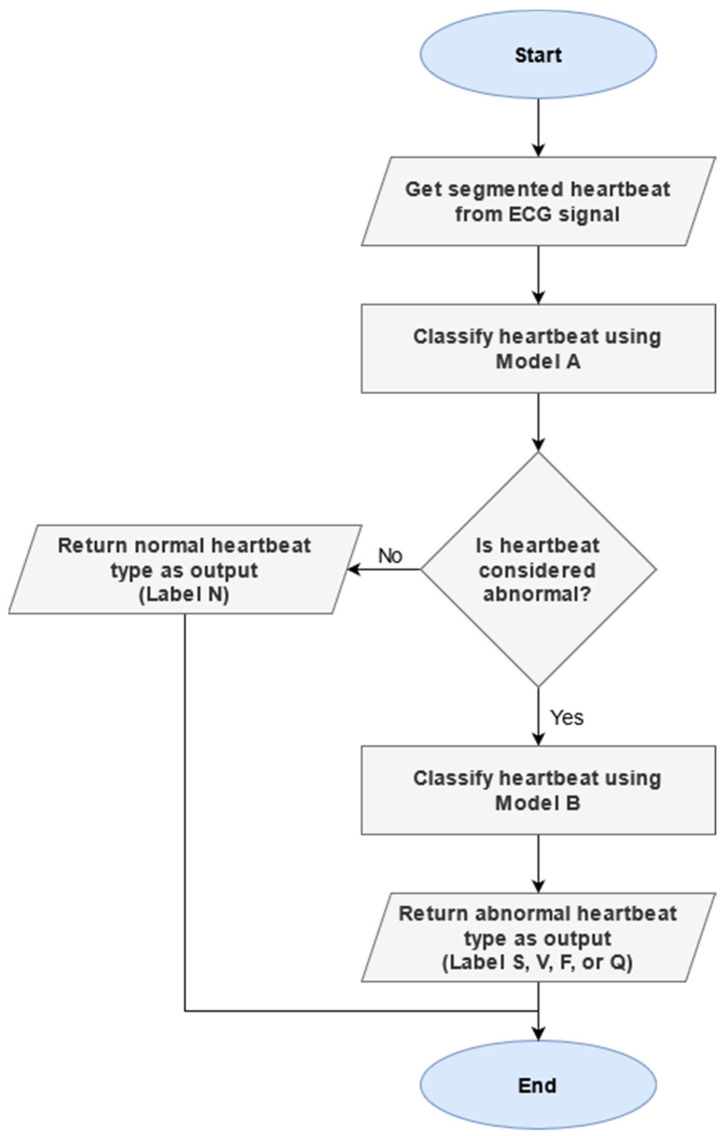
Flowchart of two-phase classification process.

**Figure 9 sensors-21-08197-f009:**
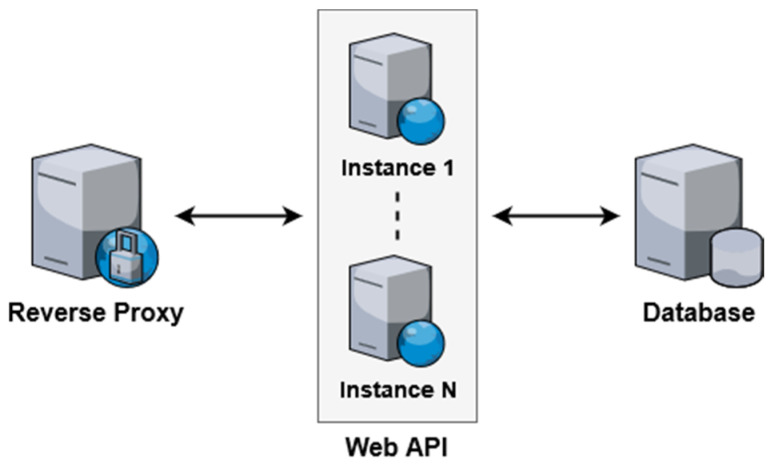
Test deployments for evaluation of horizontal scaling.

**Figure 10 sensors-21-08197-f010:**
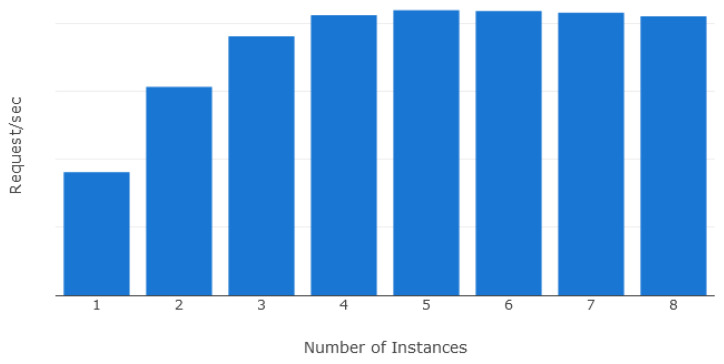
Requests/sec in relation to number of instances.

**Figure 11 sensors-21-08197-f011:**
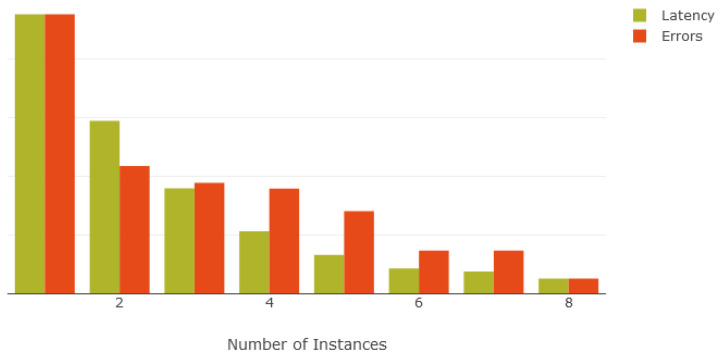
Latency and request errors in relation to number of instances.

**Figure 12 sensors-21-08197-f012:**
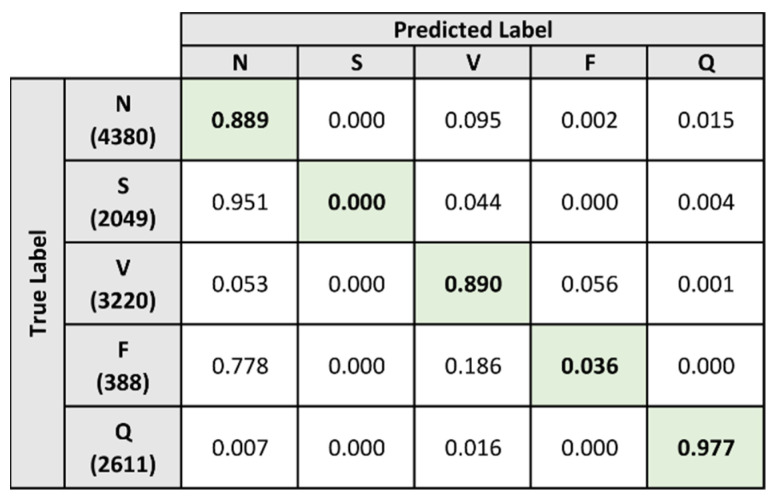
Normalized confusion matrix of baseline heartbeat classification on test dataset.

**Figure 13 sensors-21-08197-f013:**
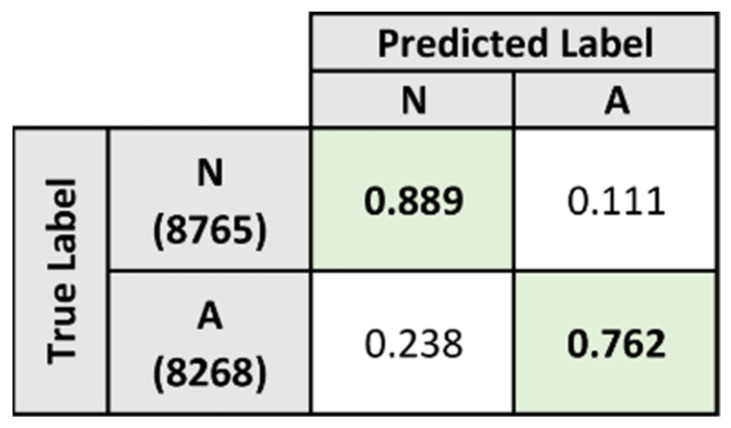
Normalized confusion matrix for Model A on test dataset.

**Figure 14 sensors-21-08197-f014:**
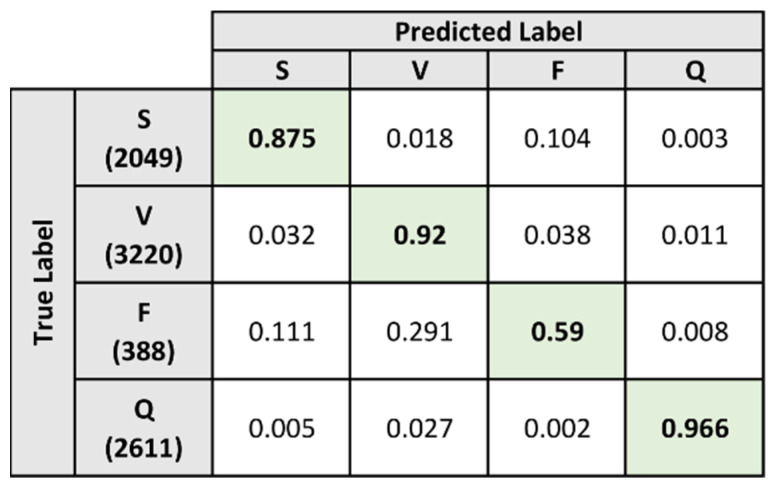
Normalized confusion matrix for Model B on test dataset.

**Figure 15 sensors-21-08197-f015:**
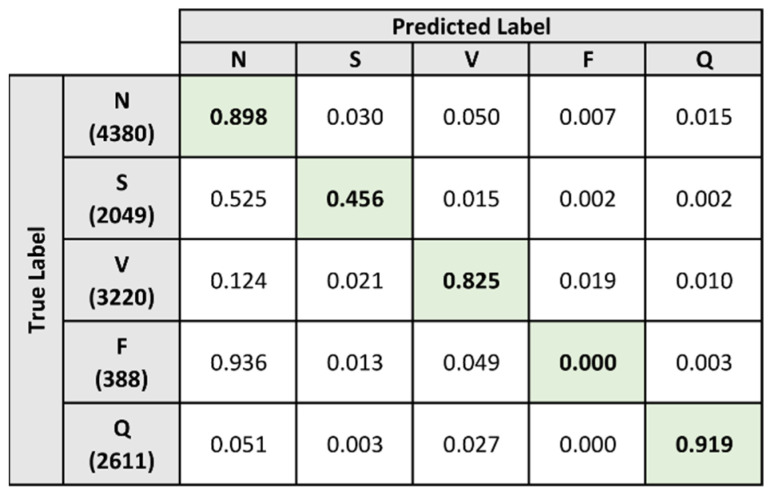
Normalized confusion matrix of baseline heartbeat classification on test dataset.

**Figure 16 sensors-21-08197-f016:**
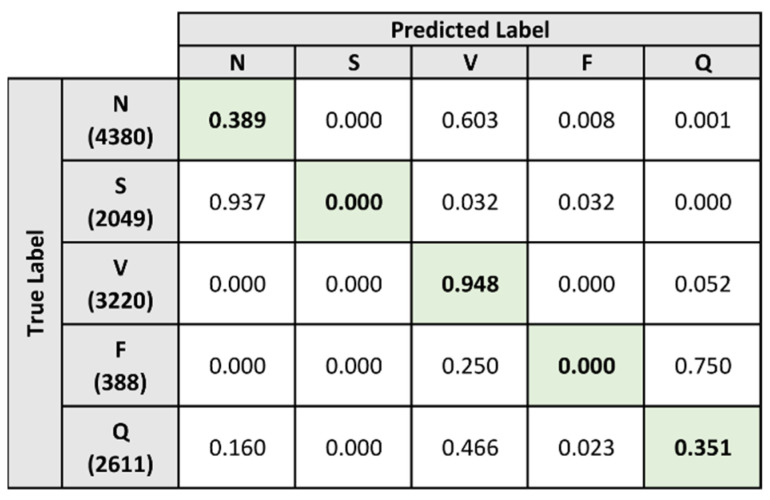
Normalized confusion matrix for baseline model on single-lead dataset.

**Figure 17 sensors-21-08197-f017:**
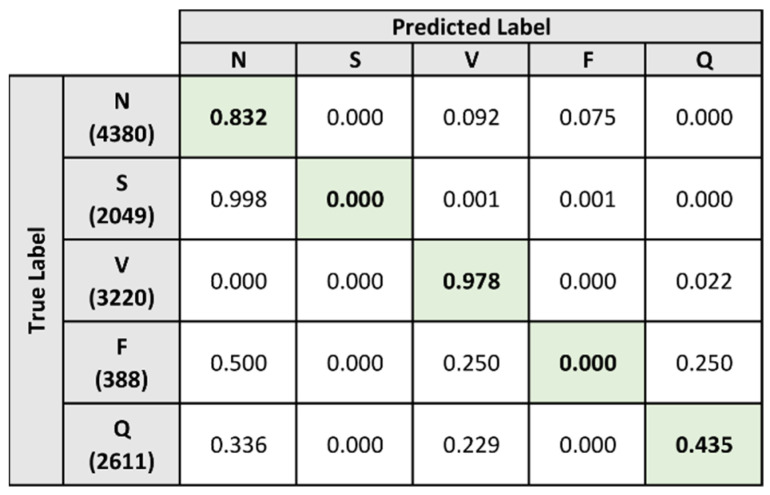
Normalized confusion matrix for composite model on single-lead dataset.

**Table 1 sensors-21-08197-t001:** Transmission interval between mobile device and server.

Transmission Interval	Description
Every 15 min	Default data transmission interval when idle
Every 5 min	Interval when actively monitoring vital signs
Every 1 min	Interval when vital signs exceed regular levels

**Table 2 sensors-21-08197-t002:** Mapping of heartbeat classification labels for ECG signals.

Original Label	Final Label	Heartbeat Type
N	N	Normal beat
L		Left bundle branch block beat
R		Right bundle branch block beat
A	S	Atrial premature beat
a		Aberrated atrial premature beat
J		Nodal (junctional) premature beat
S		Supraventricular premature or ectopic beat (atrial or nodal)
e		Atrial escape beat
j		Nodal (junctional) escape beat
V	V	Premature ventricular contraction
E		Ventricular escape beat
F	F	Fusion of ventricular and normal beat
/	Q	Paced beat
f		Fusion of paced and normal beat
Q		Unclassifiable beat

**Table 3 sensors-21-08197-t003:** Division of PhysioNet ECG records between the training and test datasets.

Record ID	Dataset Assignment
A00022	Training
A00034	Test
A00056	Training
A00106	Test
…	…

**Table 4 sensors-21-08197-t004:** Performance metrics of baseline heartbeat classification on test dataset.

Metric (%)	Overall	Labels				
		N	S	V	F	Q
Accuracy	89.481	76.866	83.799	92.307	95.533	98.901
Sensitivity	55.827	88.858	0	89.006	3.608	97.664
Specificity	92.322	70.513	100	93.435	98.442	99.223

**Table 5 sensors-21-08197-t005:** Performance metrics for Model A on test dataset.

Metric (%)	Overall
Accuracy	82.745
Sensitivity	76.173
Specificity	88.945

**Table 6 sensors-21-08197-t006:** Performance metrics for Model B on test dataset.

Metric (%)	Overall	Labels			
		S	V	F	Q
Accuracy	95.391	94.981	94.219	93.977	98.391
Sensitivity	83.775	87.506	92.019	59.021	96.553
Specificity	97.001	97.443	95.622	95.698	99.240

**Table 7 sensors-21-08197-t007:** Performance metrics of two-phase heartbeat classification on test dataset.

Metric (%)	Overall	Labels				
		N	S	V	F	Q
Accuracy	91.385	80.898	89.500	92.876	96.149	97.502
Sensitivity	61.959	89.817	45.583	82.516	0	91.881
Specificity	93.747	76.173	97.990	96.415	99.192	98.964

**Table 8 sensors-21-08197-t008:** Overall performance metrics for baseline model and composite model.

Method	Overall Metrics (%)		
	Accuracy	Sensitivity	Specificity
Baseline Model	89.841	55.827	92.322
Composite Model	91.385	61.959	93.747

**Table 9 sensors-21-08197-t009:** Number of heartbeats for each classification label.

Label	Heartbeat Count
N	18,801
S	1392
V	135
F	4
Q	131

**Table 10 sensors-21-08197-t010:** Performance metrics of baseline model on single-lead ECG signals.

Metric (%)	Overall	Labels				
		N	S	V	F	Q
Accuracy	74.622	37.350	93.198	44.036	99.047	99.477
Sensitivity	33.758	38.860	0	94.815	0	35.115
Specificity	72.587	20.277	100	43.698	99.066	99.892

**Table 11 sensors-21-08197-t011:** Performance metrics of composite model on single-lead ECG signals.

Metric (%)	Overall	Labels				
		N	S	V	F	Q
Accuracy	90.948	77.555	93.198	91.360	93.051	99.575
Sensitivity	44.898	83.203	0	97.778	0	43.512
Specificity	79.596	13.658	100	91.317	93.069	99.936

**Table 12 sensors-21-08197-t012:** Overall performance metrics for baseline model and composite model.

Method	Overall Metrics (%)		
	Accuracy	Sensitivity	Specificity
Baseline Model	74.622	33.758	72.587
Composite Model	90.948	44.898	79.596

## Data Availability

All records and data are to be kept strictly CONFIDENTIAL and can only be used for the purpose of this study.

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
