# Peer review of "Remote Arrhythmia Detection for Eldercare in Malaysia"

_sensors, 2021, doi:10.3390/s21248197_

Round 1

Reviewer 1 Report

The paper looks interesting and promising. In the paper, the authors present a scalable system architecture to support remote streaming of ECG signals and two-phase classification scheme to improve the performance of existing ECG classification algorithms. The results of two-phase classification were disputable compared to the baseline model from the MIT-BIH and PhysioNet databases, but they worked well in detecting arrhythmias from holters in real use.

  • The article is clear, easy to read and well formatted. I found only minor errors in text, such as “Android client” (lines 147, 159…) but in the figures they call it “Mobile Client”. Please unify the wording.
  • Is it a mobile (Android application) custom design or VivaLNK product (which also offers arrhythmia detection)? Or are you a member of VivaLNK? Please explain the boundary in the paper, what is your own research and what is commercial? If it is a custom design please add details about the Android application. All I see is that it was “developed using the Kotlin programming language” (lines 222-227).
  • The parallel architecture has its advantages but also disadvantages. Is it really that new, doesn't it normally exist? Could you list competing parallel systems? Of course, it is more modular, but on the other hand, does costs and complexity increase? Could you at least add an estimate to compare these parameters with standard systems? One paragraph is enough.
  • You have chosen the classification algorithm proposed by Mahajan et al. However, it filters all ECG recording using a band-pass filter between 5 – 26 Hz and resamples to 200 Hz. I understand that it's great to simplify calculations and lower data flows, but then it is still comparable to more demanding algorithms? Are there any outputs or comparisons available?
  • In 4.2.2. Data Preparation - why you upsampled PhysioNet data from 300 Hz to 360 Hz and removed the baseline and peaks with median filters to match the MIT-BIT records, when the Mahajan algorithm downsamples everything back to 200 Hz and the band-pass filter 5 – 26 Hz is guaranteed to remove this baseline and peaks. Was this step necessary?
  • Are the results of the confusion matrixes not related to the fact that in the databases the data are stored at a sampling frequency of 300 resp. 360 Hz and in real holter recordings do you have a sampling rate of 128 Hz? Thus, records from holters are not artificially degraded as when using databases. The use of the Mahajan algorithm is good for samples up to 200 Hz, but reliability decreases when evaluating better records. Could you please explain this point to me, or if you think it is appropriate to include it in your work?

Author Response

  1. We have unified the wording in the text, such as the “Android client”. Thank you.
  2. The Android mobile application and the rest of the system was implemented from scratch. We have clarified in the text that only the ECG patches and software development kit (SDK) for Bluetooth communication with the patches are created by VivaLNK. Thank you.
  3. The proposed architecture is designed to support both the traditional vertical scaling and horizontal scaling, but the scalability and continuous monitoring considerations are not normally considered in existing remote patient monitoring systems. Cost and complexity depend on how the system is scaled up, so we have added a paragraph comparing the two approaches. Thank you.
  4. In the original Physionet 2017 Challenge, the resampling from 300Hz to 200Hz is shown to not negatively impact the results as its performance is tied in the top 5. The ECG sampling rate only needs to be sufficient for the relevant features to be extracted. Thank you.
  5. We perform the pre-processing steps outlined in the methodology to ensure consistency in all ECG data used as input to the model. The code implementation of the model expects all input ECG data to be at one specific sampling rate, necessitating the initial resampling. The median filters are not necessary for this specific algorithm but provides consistency when applying the methodology to other algorithms with unclarified pre-processing stages in the future. Thank you.
  6. For the final evaluation, the ECG recorded at 128Hz are upsampled to 360Hz to match the previous training and testing data. As long as the original sampling frequency is sufficient to extract relevant ECG features, a higher sampling frequency does not reduce reliability. In the original paper, Mahajan downsampled the ECG recordings from 300Hz to 200Hz in order to reduce computation time during feature extraction. Thank you.

Reviewer 2 Report

In this paper, the author implemented a medical platform for remote monitoring and automated arrhythmia detection in elderly individuals. However, in my opinion, there is a serious problem in the paper: The authors used a lot of contents to describe and prove the classification algorithm proposed by Mahajan et al. [25] on the MIT-BIH ECG Arrhythmia Database. This algorithm has been proved in the original paper. This paper should pay more attention to the actual effect of the algorithm on remote arrhythmia detection for eldercare in Malaysia. In adddtion, this paper can also be improved from the following aspects: 1. The methodology only need ECG signal. Please explain why this system contains the temperature monitoring module? 2. For Fig. 1, two major sections can be framed separately. 3. Abbreviations that appear only once can be deleted. For example, WSN, WBAN, RPM. 4. The full name of the abbreviation should be given when it first appears. For exapmle, AAMI first appears in Line 105 Page 3, but the Advancement of 251 Medical Instrumentation (AAMI) appears in Line 251 Page 7.

Author Response

  1. The temperature module was developed as a part of the prototype system that was deployed for evaluation by the medical professionals. The module is included in the paper to demonstrate how the module system operates and future considerations for other ECG and non-ECG medical devices. Thank you.
  2. We have adjusted the figure to better demonstrate the two major sections. Thank you.
  3. We have removed the abbreviations that appear once and are not repeated. Thank you.
  4. We have adjusted the positions of the full names for abbreviations. Thank you.

Round 2

Reviewer 2 Report

I am satisfied with the response of the authors.